



# 1 Quantifying Greenhouse Gas Emissions from Woodfuel used in
# 2 Households

Alessandro Flammini[1,2], Hanif Adzmir[3], Kevin Karl[1,3], Francesco N. Tubiello[1]
[1] Food and Agriculture Organization, Rome, Italy
[2] United Nations Industrial Development Organization, Vienna, Austria
[3] Center on Global Energy Policy, Columbia University, New York, USA
*Corresponding author:* Alessandro Flammini, alessandro.flammini@fao.org
**Abstract.** The combustion of woodfuel for residential use is often not considered to be a source of greenhouse gas
(GHG) emissions in households since emissions from woodfuel combustion can be offset by the CO2 absorbed by
the growth of the forest as a carbon sink (IPCC, 2006). However, this only applies to wood that is harvested in a
renewable way, i.e., at a rate not exceeding the regrowth rate of the forest from which it is harvested (Drigo et al.,
2002). This paper estimates the share of GHG emissions attributable to non-renewable woodfuel harvesting for
use in residential food activities. It adds to a growing research base estimating GHG emissions from across the
entire agri-food value chain, from the manufacture of farm inputs, through food supply chains, and finally to waste
disposal (Tubiello et al., 2021). Country-level information is generated from United Nations Statistics Division
(UNSD) and International Energy Agency (IEA) data on woodfuel use in households. We find that, in 2019, annual
emissions from non-renewable woodfuel use in household food consumption were about 745 million tonnes
(Mt CO2eq yr$^{-1}$), with uncertainty ranging from −20 % to + 22 %, having increased 6% from 1990. Overall, global
trends were a result of counterbalancing effects: the emission increases were largely fuelled from countries in Sub-
Saharan Africa, Southern Asia, and Latin America while significant decreases were seen in countries in Eastern
Asia and South-eastern Asia. The Food and Agriculture Organisation of the United Nations (FAO) has developed
and regularly maintains a database covering GHG emissions from the various components of the agri-food sector,
including pre- and post-production activities, by country and world regions. The dataset is developed according to
International Panel on Climate Change guidelines (IPCC, 2006), which avoids overlaps across AFOLU and energy
components. It relies mainly on UNSD Energy Statistics data, which are used as activity data for the calculation
of the GHG emissions (Tubiello et al., 2022). The information used in this work is available as open data with
DOI https://doi.org/10.5281/zenodo.7310932 (Flammini et al., 2022).
**Keywords:** Agri-food systems, GHG emissions, sustainable woodfuel, household, food consumption



## 1. Introduction

In 2019, about 27% of the global population relied on traditional biomass (wood, crop residues, animal dung, etc) to meet household energy needs (IEA, 2020). The dependence on woodfuel is greatest in developing countries where it provides about one-third of total energy and is commonly used for cooking and residential heating (FAO, 2010). Approximately 70% of households in Sub-Saharan Africa depend on wood-based biomass as their primary cooking fuel. That figure is roughly 44% in South-East Asia (World Bank, 2011).

Woodfuel for domestic purposes is obtained from many supply sources, not only from forestlands. These sources include trees outside forests (such as scrubs, bush fallow, dead wood, dry branches, twigs), trees planted with agricultural crops (agroforestry or forest plantations), residues of wood harvesting, by-products of land cover change, and salvage harvesting (FAO, 2010). Several studies have examined the impact of woodfuel use in households on deforestation and human health. For the former, extensive research was conducted as a response to the 1970s and 1980s "fuelwood crisis", where conclusions were made that harvesting of fuelwood for energy is not the primary source of forest depletion (Arnold et al., 2006; Dewees, 1989; World Bank, 2011).

In terms of impact on human health, around 3.2 million premature deaths are caused due to the inhalation of polluted air in households, sourced mainly from the traditional use of biomass for heating and cooking. The pollution comes in the form of small particles that are absorbed into the lungs and enter the bloodstream. Air is considered polluted when the mean concentration of particulate matter ($PM_{10}$ and $PM_{2.5}$) and other combustion-derived indoor pollutants such as Carbon Monoxide are beyond WHO air quality guideline values (WHO, 2014). Another study pointed at an estimation of 3 million deaths per year from indoor air pollution by open fires and smoky stoves (IEA, 2021; WHO, 2021). However, very few studies have examined the climate impact of woodfuel consumption for residential use, except in the context of carbon offsets for carbon financing (e.g., using improved cookstoves). For example, one report estimated that the global potential for GHG emission reductions for improved cookstoves (ICS) is estimated at 1 Gt $CO_2$ per year (Lee et al., 2013).

This paper strives to quantify the GHG emissions attributable to household food systems consumption of woodfuel. Previous reports have set the $CO_2$ emissions associated with woodfuel consumed in households to 0 which is in line with International Panel on Climate Change (IPCC) guidelines (IPCC, 2006). Such emissions are in fact covered by the 'forestry' and 'land use' components of the AFOLU sector, while the limited emissions of $CH_4$ and $N_2O$ from woodfuel burning are reported under the Energy sector. This is based on two assumptions: i. combustion of biomass is considered renewable and has no net $CO_2$ emissions impact (the $CO_2$ absorbed by the tree during its growth is equivalent to the amount released during burning or decomposition process); ii. all $CO_2$ that is sequestered over the years by trees is released during burning. Therefore, the wood removed by land-cover change (net forest conversion), or forestland degradation will eventuate, at some point, into a release of $CO_2$. Following the IPCC approach, it is not possible to single out the amount of $CO_2$ associated with woodfuel burning at the household used for cooking.

In a renewable biomass harvesting scenario, the expectation is that the wood removed will fully regrow. New trees take up the carbon that is produced by the combustion the carbon balance in the atmosphere remains neutral. On the other hand, woody biomass is non-renewable if its extraction results in a long-term loss in carbon stocks, i.e., if the extraction rate does not allow the biomass to regrow (Drigo et al., 2014). At the same time, to estimate the real emissions associated with woodfuel, it is not possible to simply apply an emissions factor to the amount of

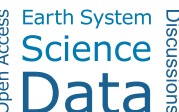

woodfuel burned, since some part of wood harvested as woodfuel can be considered sustainable. This is determined
if the rate of extraction is at or below the annual increment.
In FAOSTAT, emissions associated with the 'unsustainable' share of woodfuel burned are already covered under
emissions on 'forestland' (forest degradation) and 'net forest conversion' (deforestation) therefore, adding
emissions from woodfuel for cooking to total agri-food emissions would result in a double-counting.
In this publication, we define the amount harvested more than the annual increment as non-renewable biomass
(NRB). Obtaining accurate information about NRB fractions has historically been a challenging exercise (Lee et
al., 2013). A milestone approach on the assessment of the fractional NRB was through the use of a spatial model
called the Woodfuel Integrated Supply/Demand Overview Mapping (WISDOM) which was first applied in Mexico
by FAO in 2002 (Drigo et al., 2002) . The WISDOM model has over the years been subsequently applied to other
countries and world regions. The paper published in 2015 by Robert Bailis, Rudi Drigo, Adrian Ghilardi and Omar
Masera presents non-renewable biomass fraction (NRBf) by applying an evolution of the WISDOM model to a
number of countries across Asia, Africa and Latin America (Bailis et al., 2015).
This paper presents a methodology to apply the NRBf to woodfuel consumption used for food in the household,
based on data from the UNSD Energy Statistics database. Our methodology does not distinguish between woodfuel
emissions associated with the deforestation component and the degradation component. However, previous
research estimated that emissions from forest degradation were one-fourth of those from deforestation in 2001–
2010, and increased to one-third in more recent years (2011–2015) (Federici et al., 2015).
The results are presented consistently to FAOSTAT countries and regions, in an effort to further expand FAOSTAT
work on disseminating data on GHG emissions from agri-food systems at the country-level. Accounting for GHG
emissions across all agri-food systems activities will help researchers, policymakers, and businesses uncover novel
climate mitigation opportunities through food system interventions.
**2.  Materials and methods**
**2.1 Gap filling**
The UNSD fuelwood data used herein are gap filled to improve the quality of the available timeseries and to
estimate data for missing countries. Notably, the original UNSD energy dataset had missing data for China for the
entire time series, and this gap was filled by complementing it with IEA Energy data for primary solid biofuels
(defined as any plant matter used directly as fuel or converted into other forms before combustion). The NRBf data
was available for most countries in regions such as Africa, Asia, and Latin America. For countries with no data
available on their NRBf values, sub-regional and regional NRBf averages were used and applied accordingly.
**2.2 Emissions estimates**
For FAO, biofuel is defined as "any fuel produced directly or indirectly from biomass", while woodfuel is
described as all types of biofuels derived directly or indirectly from woody biomass (grown on either forest or non-
forest land) (FAO, 2004). GHG emissions are calculated according to the IPCC guidelines, at Tier 1 (IPCC, 2006),
by applying the following formula:
$$E_{i,g} = A_{iy} * f_w * NRBf_i * EF_g \qquad (1)$$
where



E = GHG emissions by gas (g) in select country or region I, for select inventory year, y, kilotonnes of $CO_2$
equivalent (kt CO2e $yr^{-1}$)
A = volume of woodfuel consumed in the household (activity data) for select country or territory i, for select
inventory year y, reported in cubic metres ($m^3$)
$f_w$ = share of woodfuel used for cooking for select country y,
NRBf = non-renewable biomass fraction for select country y, based on FAO WISDOM,
EF = emission factor of woodfuel, by gas, based on IPCC (2006) default values,
The volume of woodfuel consumed in the household is extracted from UNSD Energy Statistics database (Flow
1231): Consumption by households and converted to energy by applying a representative calorific value of 11.2.
The calorific value is calculated by multiplying the average heating value of air-dried wood fuel and completely
dry wood and its average density.  This heating value is estimated from the heating value of woods typically used
as woodfuel, as reported in the IEA Energy Statistics Manual (IEA, 2004). The average density of the woodfuel is
estimated by taking the density of woods typically available in tropical countries (FAO, 2007).  This assumes that
most of wood harvesting for food preparation takes place in pan-tropical countries. The share of woodfuel used
for cooking is set to unity for all tropical countries concerned (i.e., in tropical countries all woodfuel used in the
household is for cooking), while countries with little-to-no tropical coverage would have its share set as 0.847. The
rest is used for heating (Daioglou et al., 2012; Morgan, 2011).
The NRBf fraction is obtained from the 'expected' NRBf, where suboptimal harvesting of woodfuel is assumed.
The NRBf referenced is taken from the average of the low plantation productivity variant and the high plantation
productivity variant of $NRB_{B1} + NRB_{B2}$. The latter is generated with the assumption that woodfuel users can meet
their woodfuel demand from both land cover change by-products and from other sources (Bailis et al., 2015). The
same NRBf is assumed for all years and countries reported.
The calculation was run using R software for all countries and world regions, mapping UNSD to FAOSTAT
countries for the application of subregional and regional statistics.

**2.3  Data uncertainty and limitations**
There are limitations and uncertainties associated with the estimates presented herein. First, we note that, although
we assume that all fuelwood in tropical households (and the vast majority in the non-tropical countries of the
regions concerned) is used for food preparation, the input data refer to fuelwood energy use in households without
further breakdown. Second, the underlying data on energy use have gaps, especially in China and Africa. For
countries with no data, these were imputed from the IEA Energy Database instead. Thirdly, for the underlying
NRB fractions, out of 90 countries and territories, 6 were imputed based on regional averages. The uncertainty in
the original woodfuel consumption data is much smaller for some countries than for others, depending on whether
the activity data are collected using specific surveys where a sense of the uncertainty can be measured, or whether
national statistical offices use proxies and/or assumptions. In our case, using the level of uncertainty for stationary
non-energy intensive industries and "well-developed statistical systems" (such as energy statistics), an uncertainty
of ±5 % can be assumed for activity data (IPCC, 2006, volume 2, chap. 2, Table 2.6; IPCC, Estimating



Uncertainties in GHG Emissions from Fuel Combustion, Table 3; Flammini et al., 2022). Uncertainty in activity
data was then combined with uncertainty in fuel emission factors (−15 % to 18 %), computed by taking the IPCC
lower and upper values of emissions factors of wood/wood waste. Uncertainty on the conversion factor is
calculated as $\pm$12%. Lastly, the uncertainty for the NRBf values was computed to be $\pm$1.3%. The resulting overall
uncertainty from the energy statistics and emission factors was obtained by applying the IPCC (IPCC, 2006) default
error propagation method, resulting in the range −20 % to + 22 %.
An additional limitation of this methodology is that, although unsustainable woodfuel extraction could be
associated with both with deforestation and forest degradation, our methodology does not single out the emissions
that are attributable to each of them.

## 3 Results

The results show that, globally, for the year 2019, the GHG emissions associated with the unsustainable (or non-
renewable) fraction of woodfuel used in households were 741,652 kt for $CO_2$ emissions, 1,987 kt for $CH_4$ and 26.5
kt for $N_2O$. Therefore, the $CO_2$eq emissions were above 0.7 Gt in 2019, 6% greater than in 1990. These emissions
can be compared with the total 4.84 Gt $CO_2$ yr$^{-1}$ from deforestation (4.04 Gt $CO_2$) and forest degradation (0.80 Gt
$CO_2$) estimated at global level by Federici et al. (2015) over 1991-2015. This amount of emissions associated with
unsustainable woodfuel use in the household should be added to the 1.3 Gt $CO_2$eq yr$^{-1}$ associated with household
food consumption (excluding bioenergy) in the year 2019 reported in FAOSTAT (Tubiello et al., 2022). Therefore
2 billion tonnes are a more precise figure of the emissions associated with human activity at this important step of
the agri-food chain.
For comparison, the GHG emissions of the whole agri-food sector amount to 16 Gt $CO_2$eq yr$^{-1}$ and 6 Gt $CO_2$eq
yr$^{-1}$ alone from post-agricultural production activities (including food processing, transport, retail and household
consumption). Household woodfuel emissions correspond to 4.7% and 12.5% respectively.
The top 10 countries (out of 90 countries covered by the dataset) are responsible for 69% of global GHG emissions
attributable to woodfuel use for household food systems in 2019. No country from Latin America and the
Caribbean were among the top 10 GHG emitters. However, in terms of GHG emission per person (based on
population data from FAOSTAT), higher values can be seen in African countries: out of the 10 top emitters, five
are from Sub-Saharan Africa, three are from Southern Asia, one from Eastern Asia and one from South-eastern
Asia.
Nigeria and India were the largest emitters in 2019 in absolute terms. It is to note that China saw the biggest
reductions in $CO_2$ emissions over the time frame of 1990 to 2019, and China was the highest emitter from 1990 to
32 2006.

Sub-Saharan Africa, South Asia and Eastern Asia were the largest emitters among subregions, although with
different trends over 2005-2019. Eastern Asia decreased over the whole period, from 138 Mt $CO_2$eq yr$^{-1}$ in 1990
to 135 Mt $CO_2$eq yr$^{-1}$ in 2000 and further decreased to 51 Mt $CO_2$eq yr$^{-1}$ in 2019, while emissions in Sub-Saharan
Africa nearly doubled from 202 in 1990 to 379 Mt $CO_2$eq yr$^{-1}$ in 2019. Southern Asia was a significant emission
source in 2019 but has increased only slightly (around 3% over a 19-year period) since 1990, from 197 to 203





Mt CO$_2$eq yr$^{-1}$ in 2019. Emissions increased only by 1% in Latin America and South-eastern Asia decreased by
more than 60% over the same period.
We also compared estimates of emissions from woodfuel use in household food consumption with the estimates
from net forestland conversion in FAOSTAT. As discussed in the Materials and methods section, FAO estimates
of emissions from net forestland conversion are proxies for deforestation emissions data. It is also important to
note that there are various sources of woodfuel use in households as described in the Introduction section, and net
forestland conversion is just one of them. On a global scale, woodfuel household food CO$_2$ emissions were between
15% to 23% of the global net forest conversion CO$_2$ emissions.
**4   Discussion**
36.3% of overall household food emissions can be attributed to unsustainable harvest of woodfuel used in the
household for cooking. Although these GHG emissions are covered in the AFOLU section, according to the IPCC
guidelines, as part of the 'deforestation' activity, these emissions are strictly related to the cooking, which happens
towards the end of the agri-food chain. This paper presents an estimation of the emissions from 'unsustainable'
woodfuel use for cooking in the households. It is important to understand the magnitude of these emissions versus
total deforestation emissions and total household emissions because any mitigation action of these emissions
cannot be enacted without addressing cooking systems. In other words, to reduce this important share of agri-food
system emissions, any mitigation action should focus on, or at least consider, providing alternative and/or more
efficient cookstoves to the users of unsustainable woodfuel for cooking. An intervention aimed only at halting
deforestation or reducing household emissions will be partial or ineffective.
The high proportion of non-renewable woodfuel consumption in regions such as Sub-saharan Africa is reflective
of the population where low-income households have a higher dependency on biomass for their energy needs
(Dutschke et al., 2006) and energy use is less varied than their middle- and upper- income counter parts (the only
two primary services are cooking and lighting) (Sovacool, 2011). The massive reduction in non-renewable
woodfuel emissions from China over the period can be attributed to the exponential income of rural farmers with
strong rural energy policies which supported the development of other energy sources (most notably, electricity)
(Yao et al., 2012).
The updated assessment of total agri-food system emissions as supplemented by the data in this work still reaffirms
previous findings and works by the IPCC (2019), Crippa et al. (2021a, b) and Tubiello et al. (2022). However, the
most significant difference with previous work was observed in relation to household consumption emissions. Our
updated value estimates were bigger than our previous estimates of 1.2 Gt CO$_2$eq. fuelled mostly by woodfuel
combustion in household food systems. FAOSTAT estimates in this work, 1.9 Gt CO$_2$eq., were more than 4 times
those of EDGAR-FOOD (with reference to 2015, the last year for which EDGAR data were available).
A notable trend with the incorporation of non-renewable woodfuel emissions into the overall household food
system emissions is the amplification of country-level emissions in countries/territories with high dependence on
woodfuel as their source of energy (Schilmann et al., 2021; World Bank, 2011). Our refined assessments of
emissions contributions highlight the importance of non-renewable woodfuel into the overall food systems
emissions. Regarding the three major components of the food system (on-farm production, land use change and



pre- and post-production activities), our analysis highlights that in 2019, pre- and post-production emissions have almost the same share of emission contributions to farm-gate activities (38.0% vs 41.7%) at the global level, while land use change provided a smaller contribution (20.3%). For the same year, household food systems took the biggest share of pre- and post-production emissions (31.5%) while non-renewable woodfuel combustion was 36.3% of household food systems.

## 5  Data availability

The GHG emission data presented herein cover the period 1990–2019 at the country level. They are available as open data, with DOI https://doi.org/10.5281/zenodo.7310932 (Flammini et al., 2022).

## 6  Conclusions

This paper provides updated details of the FAOSTAT database on GHG emissions along the entire agri-food systems chain (Tubiello et al., 2022), with a focus on improving the estimates of the household consumption emissions.

The data are provided in open-access mode to users worldwide and are available by country over the period 1990-2019, with plans for annual updates. The major trends in non-renewable woodfuel consumption within household food-systems that were identified in this work can help locate emissions hotspots in agri-food systems and inform the adoption/effectiveness of policies on cooking fuel switches on the country, regional and global level. This work also emphasizes the increasingly important role that pre- and post-production processes along supply chains play in the overall GHG footprint of agri-food systems, in a regional and global level.

This paper also helps to expand the impacts of woodfuel use beyond just health measures but to also highlight the climate impact attached to using non-renewable woodfuel as a source of cooking fuel. Finally, the methodological work underlying these efforts complements and extends recent pioneering efforts by FAO and other groups in characterizing technical coefficients to enable quantifying the weight of agri-food systems within countries' emissions profiles.

## 7  Competing interests

At least one of the (co-)authors is a member of the editorial board of *Earth System Science Data*. The peer-review process was guided by an independent editor, and the authors also have no other competing interests to declare.

## 8  Disclaimer

The views expressed in this paper are the authors' only and do not necessarily reflect those of FAO or UNIDO.

## 9  Acknowledgements



FAOSTAT is supported by the FAO regular budget, funded by its member countries. We acknowledge the efforts of national experts who provide the statistics on energy use that are at the basis of this effort. All authors contributed critically to the drafts and gave final approval for the publication. We are grateful for overall support by the Food Climate Partnership at Columbia University.

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



TABLES
**Table 1**. Typical heating values of woods used as fuelwood

| Wood Type | Heating value |
|---|---|
| Air-dried wood (10% to 20% moisture content) | 16 MJ/kg |
| Completely dry wood (oven-dried) | 18 MJ/kg |
| Average | 17 MJ/kg |

Source: IEA Biofuel Energy Statistics (Page 174 of the IEA Energy Statistics Manual) (IEA, 2004)
**Table 2**. Typical densities of woods used as fuelwood

| Wood Type | Density |
|---|---|
| Air-dried wood | 725 kg/m$^3$ |
| Oven-dried wood | 593 kg/m$^3$ |
| Average | 659 kg/m$^3$ |

Source: Wood-energy supply/demand scenarios in the context of poverty mapping (Table A2.4) (FAO, 2007)
**Table 3.** List of tropical countries

| | | | |
|---|---|---|---|
| Algeria | Benin | Lao People's Democratic Republic | Guinea-Bissau |
| Angola | Dominica | Liberia | Timor-Leste |
| Antigua and Barbuda | Dominican Republic | Libya | Rwanda |
| Argentina | Ecuador | Madagascar | Saint Helena |
| Bolivia (Plurinational State of) | El Salvador | Malawi | Saint Kitts and Nevis |
| Botswana | Equatorial Guinea | Malaysia | Anguilla |
| Brazil | Ethiopia | Maldives | Saint Lucia |
| Belize | Eritrea | Mali | Saint Vincent and the Grenadines |
| Solomon Islands | Falkland Islands (Malvinas) | Mauritania | Sao Tome and Principe |

| | | | |
|---|---|---|---|
| British Virgin Islands | Fiji | Mauritius | Senegal |
| Brunei Darussalam | French Guiana | Mexico | Seychelles |
| Myanmar | Djibouti | Mozambique | Sierra Leone |
| Burundi | Gabon | Namibia | Viet Nam |
| Cambodia | Gambia | Nepal | Somalia |
| Cameroon | Ghana | Curacao | South Africa |
| Cabo Verde | Grenada | Aruba | Zimbabwe |
| Central African Republic | Guadeloupe | Saint Maarten (Dutch part) | South Sudan |
| Sri Lanka | Guatemala | Bonaire, Sint Eustatius and Saba | Sudan |
| Chad | Guinea | New Caledonia | Suriname |
| Chile | Guyana | Vanuatu | Togo |
| Colombia | Haiti | Nicaragua | Trinidad and Tobago |
| Comoros | Honduras | Niger | Turks and Caicos Islands |
| Mayotte | India | Nigeria | Uganda |
| Congo | Indonesia | Panama | Egypt |
| Democratic Republic of the Congo | Cote d'Ivoire | Papua New Guinea | United Republic of Tanzania |
| Costa Rica | Jamaica | Paraguay | Venezuela (Bolivarian Republic of) |
| Cuba | Kenya | Peru | Guinea-Bissau |

1   Source: Journal of Tropical Psychology, Volume 1 (Morgan, 2011)

3   **Table 4.** Non-renewable fractions (NRBf) based on country and region

| Country/region | NRB fraction | Country/region | NRB fraction |
|---|---|---|---|
| Angola | 0.350 | Malawi | 0.371 |
| Argentina | 0.283 | Malaysia | 0.465 |
| Bangladesh | 0.510 | Mali | 0.291 |
| Belize | 0.993 | Mauritania | 0.348 |
| Benin | 0.217 | Mexico | 0.268 |





| | | | |
|---|---|---|---|
| Bhutan | 0.559 | Mozambique | 0.397 |
| Bolivia (Plurinational State of) | 0.325 | Myanmar | 0.085 |
| Botswana | 0.895 | Namibia | 0.476 |
| Brazil | 0.238 | Nepal | 0.524 |
| Brunei Darussalam | 0.872 | Nicaragua | 0.579 |
| Burkina Faso | 0.476 | Niger | 0.235 |
| Burundi | 0.570 | Nigeria | 0.511 |
| Cambodia | 0.384 | Pakistan | 0.836 |
| Cameroon | 0.758 | Panama | 0.496 |
| Central African Republic | 0.264 | Papua New Guinea | 0.403 |
| Chad | 0.237 | Paraguay | 0.384 |
| Chile | 0.138 | Peru | 0.309 |
| China | 0.16 | Philippines | 0.214 |
| Colombia | 0.344 | Rwanda | 0.585 |
| Congo | 0.099 | Senegal | 0.361 |
| Costa Rica | 0.18 | Sierra Leone | 0.219 |
| Côte d'Ivoire | 0.163 | Singapore | 0.755 |
| Democratic Republic of the Congo | 0.24 | Solomon Islands | 1 |
| Dominican Republic | 0.33 | Somalia | 0.524 |
| Ecuador | 0.99 | South Africa | 0.238 |
| El Salvador | 0.372 | Sri Lanka | 0.244 |
| Equatorial Guinea | 0.94 | Sudan | 0.411 |
| Eritrea | 0.679 | Suriname | 0.181 |
| Ethiopia | 0.613 | Thailand | 0.03 |
| French Guiana | 0.165 | Timor-Leste | 1 |
| Gambia | 0.412 | Togo | 0.44 |
| Ghana | 0.286 | Trinidad and Tobago | 0.554 |
| Guatemala | 0.334 | Uganda | 0.613 |





| | | United Republic of Tanzania | 0.235 |
|---|---|---|---|
| Guinea | 0.297 | | |
| Guinea-Bissau | 0.279 | Venezuela (Bolivarian Republic of) | 0.527 |
| Guyana | 0.039 | Viet Nam | 0.115 |
| Haiti | 0.666 | Zambia | 0.340 |
| Honduras | 0.637 | Zimbabwe | 0.377 |
| India | 0.231 | Eastern Asia | 0.16 |
| Indonesia | 0.434 | Latin America and the Caribbean | 0.396 |
| Jamaica | 0.185 | Melanesia | 0.702 |
| Kenya | 0.635 | Northern Africa | 0.369 |
| Lao People's Democratic Republic | 0.273 | South-eastern Asia | 0.421 |
| Lesotho | 0.525 | Southern Asia | 0.484 |
| Liberia | 0.283 | Sub-Saharan Africa | 0.415 |
| Libya | 0.327 | | |

1    *Source: Authors' own elaboration.*





FIGURE LEGENDS
**Figure 1.** Global GHG emissions from woodfuel use in households for cooking from 1990 to 2019 (Mt), including
uncertainty ranges. Source: Authors, based on data from IEA and UNSD (2022)
**Figure 2.** GHG emissions trends from the top 10 emitters of 2019 from 1990 to 2019 (Mt CO2eq). Source: Authors.
**Figure 3.** Share of global GHG emissions from household woodfuel use in food sector stratified according to sub-
region. Source: Authors.
**Figure 4.** Trends in $CO_2$ emissions from net forestland conversion and woodfuel use in household food
consumption from 1990 to 2019. Source: FAOSTAT, 2022
**Figure 5**. Proportion of emissions for non-renewable woodfuel use in household food systems in comparison to i)
the overall food system (Pie Chart 1), ii) pre- and post-production (Pie Chart 2) and iii) household food systems
(Pie Chart 3), for the year 2019. Source: FAOSTAT, 2022

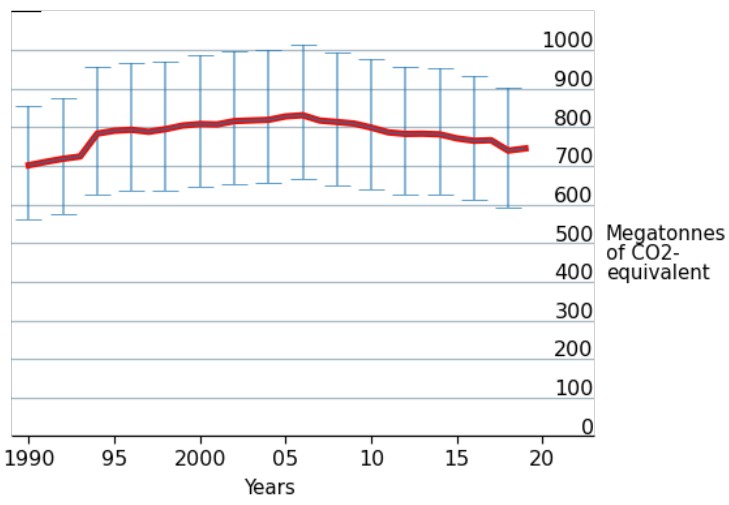

**Figure 1.** Global GHG emissions from woodfuel use in households for cooking from 1990 to 2019 (Mt), including uncertainty ranges. Source: Authors, based on data from IEA and UNSD (2022)

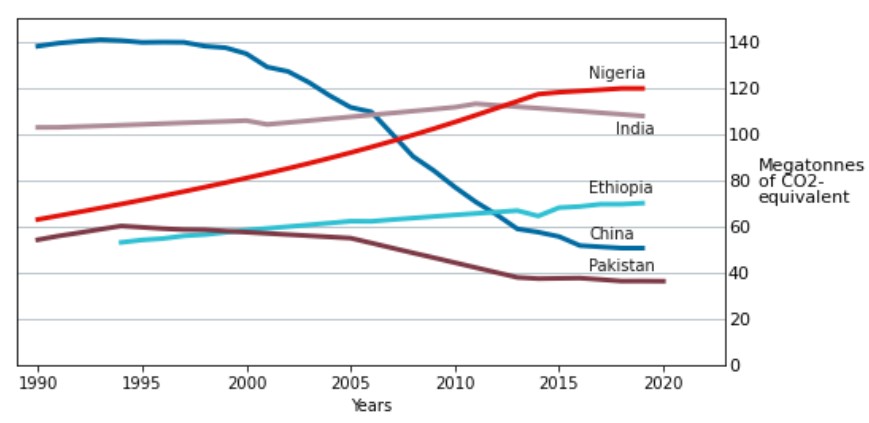

**Figure 2.** GHG emissions trends from the top 10 emitters of 2019 from 1990 to 2019 (Mt CO2eq). Source: Authors.

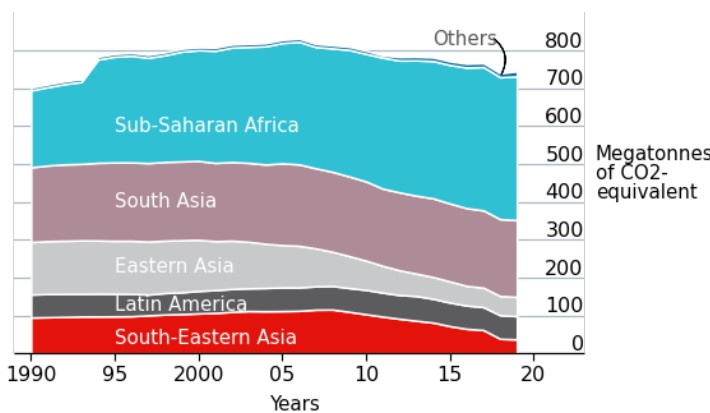

**Figure 3.** Share of global GHG emissions from household woodfuel use in food sector stratified according to sub-region.
Source: Authors.

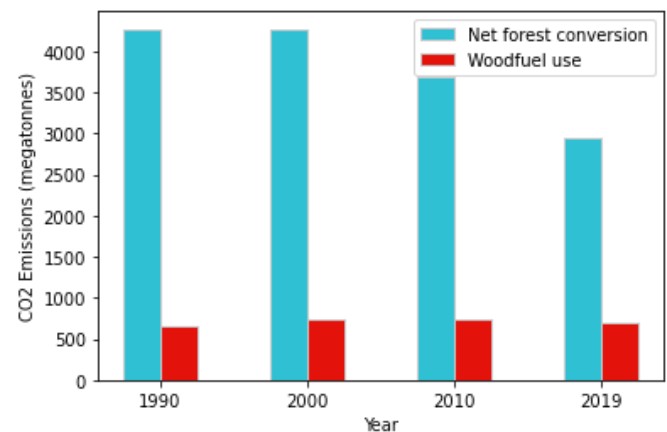

**Figure 4.** Trends in CO2 emissions from net forestland conversion and woodfuel use in household food consumption from
1990 to 2019. Source: FAOSTAT, 2022

*Food Systems*

*Pie Chart 1*

*Pre- and Post Production*

*Pie Chart 2*


*Household Food Systems*


*Pie Chart 3*

**Figure 5.** Proportion of emissions for non-renewable woodfuel use in household food systems in comparison to the overall food system (Pie Chart 1), pre- and post-production (Pie Chart 2) and household food systems (Pie Chart 3) for the year 2019. Source: FAOSTAT, 2022


