# Peer review of "Quantifying Greenhouse Gas Emissions from Woodfuel used in"

_Earth System Science Data, 2022_

## Author Response (AR1)

Thank you for the thorough review. Hopefully we addressed all points raised by the reviewers.

Anonymous referee #1

Major concerns

1. The most important parameter to calculate the non-renewable biomass consumption is NRBf (non-renewable biomass fraction). It seems that NRBf values were directly borrowed from previous work. What is the contribution of this work? Are there any updates on NRBf? These should be made clearer.

   Response: The NRBf does constitute a key parameter in our calculations. In Bailis 2015, the NRBf values are disaggregated into spatial regions, scenarios, and their productivity variant. The data on the volume of woodfuel consumed in the UNSD Energy Statistics Database does not distinguish between sustainable and non-sustainable biomass, thus our work aims to provide a working methodology to apply the NRBf values to the UNSD energy statistics to obtain country-level emission statistics. We use NRBf values of scenarios that best reflect the country's situation. This paper works as a component of "information items" (IPCC 2006, Chapter 2, 2.3.3.4) to enable users to have a better understanding of emissions attributable to woodfuel use in households, as well as to make country, regional and global statistics of woodfuel associated emissions. As far as we know, this is the first attempt to quantify emissions from non-sustainable wood used in households by country and with global coverage. The data will eventually be used to improve the FAOSTAT datasets.

2. Still for NRBf, are there temporal changes in NRBf? The non-renewable fraction of biomass should vary over time. Assuming consistent values for individual places may lead to temporal bias, which should be addressed.

   Response: There is a temporal bias attached to this paper as all NRBf values are obtained with 2009 being used as a base year. Such bias is now clarified in the paper and will be addressed in our future research.

3. To improve the quality of biomass burning activity data, the UNSD fuelwood data were gap filled using IEA's solid biofuel data. However, both sets of data for biofuel statistics are subject to high uncertainty. The authors assumed an uncertainty of ±5%, which adopted the uncertainty level considered for "stationary non-energy intensive industries" or "well-developed statistical systems" in IPCC (lines 35-37, page 4). However, household biomass burning doesn't belong to "stationary non-energy intensive industries" or "well-developed statistical systems". The biomass statistics should have uncertainty much larger than 5%. We typically assume a CV of 20% for biomass. The uncertainty of the work is likely underestimated and should be reconducted.

   Response: Thank you for pointing this out. A reassessment of the uncertainty has been re-conducted and the following conclusions are made: i. the use of UNSD fuelwood data and IEA solid biofuel data are categorized as "Less developed statistical systems" ii. The sector of emissions uncertainty is now categorized as "Biomass in small sources". Such categorizations put the uncertainty at ±60%.

Specific comments,

1. Line 24, page 2, the term "household food system" should be defined before being used.
   DONE.

2. Lines 21-22, page 4, "The same NRBf is assumed for all years and countries reported." Does it mean all years and countries use the same NRBf value? It is clearly not the case.
   DONE, rephrased.

3. Line 1, page 5, there are two "Flammini et al., 2022" in the reference list. Please distinguish in the main text.

   DONE.

4. Line 8, page 5, remove the second "with"

   DONE

5. Line 31, page 6, "1.2 Gt CO2eq": It is not very clear what this value stands for. Does it stand for the total GHG emissions from household energy consumption or something else? In addition, this sentence needs reference.

   DONE, and reference added.

6. Line 33, page 6, ""EDGAR-FOOD". reference is needed.

   DONE, and reference added.

7. Line 1, page 7, the terms "pre- and post-production" were brought up here all of a sudden. They should be defined in the method section. How their emissions were estimated should be provided.

   Rephrased.

8. Content in the discussion section should be based on the content in the result section. However, it seems that many of the current content in the discussion section were based on something else that were not mentioned in the result section, nor in the method section.

   This comment is not very specific. However, we moved the paragraph comparing household food emissions to total food emissions to the Discussion. We kept the brief description on how the results compare with other literature results under Results. We hope the revised version of the manuscript addresses the concern of the reviewer.

Anonymous referee #2

Quantification of greenhouse gas (GHG) emissions is globally one of the most significant research points. This study presented global emissions of GHG from woodfuel used in households. These emission data could be valuable for use in climate models to further evaluate impact of such emission impact on climate. However, it seems that this present paper was not well-prepared. One of my major concerns is that the author should clearly address the contribution for building such emission data set. As reported, many data, parameters, and methodology seem taken from previous studies/reports. In addition, the uncertainty for this

emission data set needs to be further evaluated in more details. For text (including introduction, results and discussion sections), they also need to be further improved. Many paragraphs were made in the paper, yet some of which seem inappropriate. Lastly, the figures presented in this paper were not well prepared neither.

For example:

Page 3, lines 6-18: the authors summarized the NBRf method, however it wasn't clearly prepared. It's hard to follow the objectives of these description.

Some additional context for this work was added.

Page 3, lines 11-12. It seems that this way, listing all names of scientists, should not be recommended normally?

Edited to remove scientists names.

Materials and methods: the authors need to clearly address their contributions to the methodology and/or data for the emissions.

We added some clarification text.

Data uncertainty and limitations: more detailed information along with supporting data or results will be further needed for data uncertainty discussion. Moreover, many numbers for the uncertainty seen directly taken from IPCC.

Since this paper uses the IPCC methodology to calculate emissions, also the uncertainty information associated is taken as much as possible from IPCC. We changed the uncertainty ranges as per suggestions from Reviewer 1.

Results and discussion sections: it seems no discussion reading figures presented in the paper? One more, it's hard to follow the relations between paragraphs, sometimes. And the discussion seems not associated with the results correspondingly. In addition, was the gridded emission data available?

All figures are referenced in the text. They are provided separately because this is the format requested by ESSD. This comment is a bit general and not clear to understand which specific changes are requested. We hope this revised version addresses the concerns. Are 'gridded emission data' the country-specific grid emission factors? These are not used for the calculation of emissions from woodfuels. The IEA grid emission factors are used for the calculation of household emissions and other food system emissions (in Tubiello et al. 2021, 2022, and in Flammini et al. 2022b) but they are not reported publicly since IEA does not make them publicly available.

Tables: table might not need to be presented in main text, while it might be recommended to be removed into supplement.

All parameters that are used directly in our methodology are reported separately at the end of the paper, as per ESSD guidelines.

Figures: the figures appear not uniform, which should be modified and standardized. In addition to trends in emissions, the emissions should be also presented in a 3-D format for better visualization of spatiotemporal distributions.

We include temporal series (trends) but the comment on the "spatiotemporal distribution" is not clear to us. We would need more specific guidance on what to report on the different axis of a 3D graph, if necessary.  If it is just an editorial edit, the journal can do it. We would be happy to improve the figures if still needed, but we would need some additional guidance.

---

## Author Response (AR2)

24 April 2023

Never offer quantification of food system emissions. Mentioned multiple times but never quantified. Thus, using Fig 5 for example, one could check (rounding) 1/3 of total food emissions (top pie chart), then 1/3 of pre- and post-production (middle chart), then finally 1/3 of household for NRB fraction. (If authors quote plus/minus 60% uncertainties, how can they then show percentages to 0.x%?) We never get any numbers to work with? Why? Working from earlier paper by same group (https://doi.org/10.5194/essd-14-1795-2022) one finds 16.5 Gt CO2eq yr (2019) for total food systems and 5.8 for pre- and post-production. 1/3 of 5.8 gives 1.9 for household emissions, times another 1/3 for NRB (authors say 36.3%) gives 0.6 -0.7 Gt, consistent with and within uncertainty range of 0.7 Gt quoted by authors. Why did I have to do this? Authors need to provide numbers! Readers should not need to chase! Show us that you provide reliable information on NRB sources!

The reviewer is right and his/her calculations also right. We agree that readers may be interested in numbers describing how household woodfuel emissions relate to agrifood system emissions, and we added them in the text.

And, clarify in Figure 1? Do these numbers represent total wood fuel use or non-renewable only. From calculations above this reader concludes total rather than NRB, but manuscript focuses on NRB. Also, for Fig 1, authors claim 6% increase 1990 to 2019 but with uncertainty ranges so large how can they certify any trends?

These numbers represent non-renewable emissions, which correspond to wood fuel emissions too, since the "renewable fraction" is assumed to be "net-zero".

Figure 2 claims "top 10 emitters" but only shows five countries?

Thank you for spotting this. It was our mistake. Now corrected.

Fig 3 supposedly shows emissions by region but, earlier (p 4, lines 14-15 and 24) authors claim justification for omitting tropical regions and fail to provide details on how they identified regions or subregions. This reader finds no justification to assess Fig 3. If NRBf declined in four regions, then emissions must have increased in the remaining one or in others not accounted?

We clarified it in the text: that regions and sub-regions are identified based on FAOSTAT categorization, and clarified which subregions are related to pan-tropical. One important assumption is that most of unsustainable wood harvesting for food preparation takes place in pan-tropical countries (i.e. in the regions represented in figure 3). Other non-tropical regions (e.g. Europe) are not covered by the analysis, since the NRBf is expected to be much lower or negligible.

Figure 4 needs error bars. Again, this reader doubts that data support any conclusion about temporal trends.

We agree that the uncertainty does not support any straight conclusion, however we can identify a trend. Error bars have been added to figure 5 as suggested.

Fig 5 addressed above, needs quantification.

Resolved (first point above).

AFOLU acronym used several times but never defined.

Good point. The acronym has been now explained in the text.